Medical Imaging with Deep Learning 2024

# DyABD: A Dataset and Technique for Synthetically Generating Dynamic Abdominal MRIs with Dual Class and Anatomically Conditioned Diffusion Models

**Niamh Belton**[1,2]                                    NIAMH.BELTON@UCDCONNECT.IE
**Victoria Joppin**[4,5]                                  VICTORIA.JOPPIN@UNIV-EIFFEL.FR
**Aonghus Lawlor** [3,6]                                 AONGHUS.LAWLOR@UCD.IE
**Kathleen M. Curran**[1,2]                              KATHLEEN.CURRAN@UCD.IE
**Catherine Masson**[4]                                  CATHERINE.MASSON@UNIV-EIFFEL.FR
**Thierry Bege**[4,7]                                    THIERRY.BEGE@AP-HM.FR
**David Bendahan**[5]                                    DAVID.BENDAHAN@UNIV-AMU.FR

[1] *Science Foundation Ireland Centre for Research Training in Machine Learning*

[2] *School of Medicine,* [3] *School of Computer Science, University College Dublin*

[4] *Aix Marseille Université, Université Gustave Eiffel, LBA, Marseille France*

[5] *Aix Marseille Université, CNRS, CRMBM UMR 7339, Marseille France*

[6] *Insight Centre for Data Analytics, University College Dublin, Dublin, Ireland*

[7]*Aix Marseille Université, Département de Chirurgie, Hôpital Nord, APHM, Marseille, France*

## Abstract

An abdominal hernia is a protrusion of intestine or tissue in the abdominal wall and is known to cause debilitating pain. The recurrence rate of abdominal hernia varies from 30% to 80%, meaning it is paramount to improve our understanding of the mechanical functionality and physiology of the abdominal wall. This work proposes DyABD, a dataset of dynamic abdominal MRIs (2D+t) of hernia patients and a 3D dual class and anatomically conditioned Denoising Diffusion Probabilistic Model (DDPM) that can perform the unique task of synthesising hernia patients performing any of three exercises, breathing, coughing or a Valsalva maneuver, whilst also taking into account whether the patient is pre or post corrective abdominal surgery. DyABD requires a subject prior as input which consists of the first 2D slice of the dynamic MRI sequence and the associated abdominal muscle masks of the first 2D slice which ensures anatomical correctness is preserved during synthesis. This work is based on 121 dynamic MRI volumes which will be made available for sharing as part of the complete dataset of approximately 300 volumes. The preliminary results of DyABD demonstrate its ability to model the mechanical functionality of the abdominal wall. Examples of generated volumes are made available at https://github.com/niamhbelton/DyABD/.

**Keywords:** Diffusion, Dynamic, MRI, Abdominal, Generative, Deep Learning, Medical Imaging, Image Synthesis

## 1. Intoduction

An abdominal hernia is a protrusion of viscera through a rupture between muscles in the abdominal wall (Stedman, 1920) and is known to cause debilitating discomfort. The recurrence rate of abdominal hernia post corrective surgery varies from 30% to 80% (Bhardwaj et al., 2024), meaning it is paramount to improve our understanding of the mechanical functionality and physiology of the abdominal wall. Dynamic MRI has recently been shown

to be effective in quantifying abdominal wall motion and deformation during various exercises such as breathing and muscular contraction (Jourdan et al., 2022; Miyamoto et al., 2002). Therefore, this work has acquired a unique dynamic abdominal MRI dataset, named DyABD that contains patients with abdominal hernias performing various exercises pre and post corrective surgery. However, data scarcity, missing data and other issues relating to ability of the patient to perform the exercises remain a challenge to progressing the field.

This work proposes DyABD, a dataset of dynamic abdominal MRIs of hernia patients and a 3D dual class and anatomically conditioned Denoising Diffusion Probabilistic Model (DDPM) that can perform the unique task of synthesising patients performing any of three exercises, breathing, coughing or Valsalva maneuver, whilst also taking into account whether the patient is pre or post corrective surgery. DyABD requires a subject prior as input which consists of the first 2D axial slice of the dynamic MRI sequence and the associated abdominal muscle masks which ensures anatomical correctness is preserved during synthesis.

## 2. Methods

**Dataset:** The dataset consists of 121 dynamic MRI volumes (2D+t), where each volume consists of a 2D axial slice acquired at the same position over time whilst performing three exercises, coughing, breathing and Valsalva maneuver from eight patients with abdominal hernias pre-corrective surgery and from three of the eight patients post-corrective surgery. The abdominal muscle masks for the first slice of the sequence were annotated manually.

Figure 1 outlines the DyABD architecture. A training dataset $X_{train} \in \mathbb{R}^{h \times w \times s}$ consists of volumes where $h$ and $w$ are the height and width of a 2D slice and $s$ is the number of slices in the time sequence. The dynamic volumes are resized so that $h = 128$, $w = 128$ and $s = 128$. DDPMs (Ho et al., 2020) consist of a forward diffusion and reverse diffusion process. The forward process, $q$ adds a small amount of Gaussian noise, $\epsilon \sim \mathcal{N}(0, 1)$ to the input volume, $x \in X_{train}$ for a given number of timesteps, $T$. In the reverse process, $p$, a U-Net (Ronneberger et al., 2015), denoted as $\epsilon_\theta$ is trained to predict the added noise, at each timestep, $t$. In order to model the sequential nature of the data, the U-Net from the original DDPM (Ho et al., 2020) implementation is modified to have a 3D U-Net architecture (Dorjsembe et al., 2022). A recent work, Med-DDPM (Dorjsembe et al., 2023) demonstrated how concatenating a conditional image with the MRI volume, $x$ at timestep $t$, improves 3D image synthesis. This work therefore, concatenates the subject prior, $sub$ with the MRI volume, $x_t$ at each timestep $t$. The subject prior, $sub \in \mathbb{R}^{c \times h \times w}$ for each $x \in X_{train}$ has five channels, $c$, where a channel exists for the first slice in the sequence and the associated masks for each of the four abdominal muscles. The subject prior, $sub$ is stacked $s$ times along the time axis before concatenation, meaning the input to the U-Net at timestep $t$ is of shape $(c + 1) \times h \times w \times s$, denoted as $x'_t$. Two label embedding layers for the class of exercise, $l_e$ and operative stage, $l_o$ are used to class condition the model. They are added to the time embedding layers of the U-Net. The training minimises the L1 loss between the predicted noise and the actual noise, $||\epsilon - \epsilon_\theta(x_t, t, l_e, l_o)||$ across all timesteps. The model is trained with $T = 250$ (as per Dorjsembe et al. (2023) implementation) for 50,000 steps.

## 3. Results and Conclusion

The model was trained using four fold cross validation, where each training fold had an average of 91 volumes and each test fold had an average of 30 volumes. Table 1 presents the

results averaged over four test folds. The Structural Similarity Index Measure (SSIM) and Mean Squared Error (MSE) was calculated between the generated volumes and the ground truth volumes. Examples of the generated volumes are available at https://github.com/niamhbelton/DyABD/. DyABD shows promising initial results that demonstrates its ability to model the mechanical functionality of the abdomen performing each of the exercises. Future work will focus on improving the resolution.

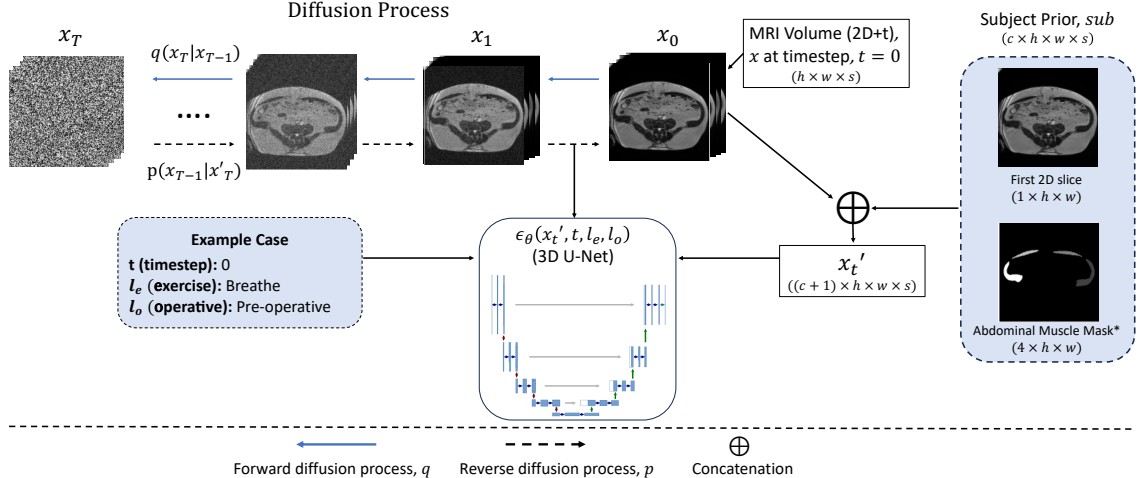

**Figure 1:** The figure shows the forward and reverse diffusion process of DyABD. In the reverse diffusion process, the U-Net model, $\epsilon_\theta$, takes $x'_t$ as input which is the concatenation of the MRI volume $x_t$ with the subject prior, $sub$. $\epsilon_\theta$ is conditioned on the timestep, $t$ and the class conditions, exercise, $(l_e)$ and operative stage $(l_o)$. *the mask of all four abdominal muscles are shown on one image for visualisation purposes.

**Table 1:** Results of four fold cross validation in terms of SSIM and MSE and one standard deviation with and without conditioning on the abdominal muscle mask (w/o mask).

| Method | Cough SSIM ↑ | Cough MSE ↓ | Breathe SSIM ↑ | Breathe MSE ↓ | Valsalva Maneuver SSIM ↑ | Valsalva Maneuver MSE ↓ | Overall SSIM ↑ | Overall MSE ↓ |
|---|---|---|---|---|---|---|---|---|
| DyABD | **0.84 ± 0.0** | **0.003 ± 0.0** | **0.71 ± 0.1** | **0.005 ± 0.0** | **0.78 ± 0.1** | **0.004 ± 0.0** | **0.78 ± 0.1** | **0.004 ± 0.0** |
| w/o mask | 0.75 ± 0.2 | 0.004 ± 0.0 | 0.58 ± 0.3 | 0.010 ± 0.0 | 0.72 ± 0.1 | 0.005 ± 0.0 | 0.69 ± 0.2 | 0.006 ± 0.0 |

Although anatomically guided diffusion models exist (Konz et al., 2024), there is a lack of models that do not require anatomical guidance for the complete volume, whilst also being class conditional and capable of modelling the sequential nature of dataset, highlighting the uniqueness of DyABD's problem set-up. To provide context to the results, we also report the performance of DyABD without conditioning on the abdominal muscle mask. The results demonstrate the importance of providing anatomical context to the model.

This work has outlined DyABD, a unique dynamic abdominal MRI dataset and technique for the novel task of synthesising hernia patients performing various exercises given a subject prior. Future work will include a qualitative study with clinicians to assess the synthesised volumes in terms of anatomical accuracy and the correctness of the functionality of the abdominal wall muscles for each exercise performed for both pre and post operative patients. The Github repository will be updated when the dataset is available for sharing.

## Acknowledgements

This work was funded by Science Foundation Ireland through the SFI Centre for Research Training in Machine Learning (Grant No. 18/CRT/6183). This work is supported by the Insight Centre for Data Analytics under Grant Number SFI/12/RC/2289 P2.

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
