# OpenReview forum: "DyABD: A Dataset and Technique for Synthetically Generating Dynamic Abdominal MRIs with Dual Class and Anatomically Conditioned Diffusion Models"
_MIDL.io/2024/Short_Papers — MIDL 2024 Short Papers_

### Official Review · Reviewer_YyEs · 2024-04-24

**Confidence:** 4
**Final Rating:** 3.5

**Review:**

This paper presents an interesting application of the diffusion model for generating abdominal MRI. Both of the method description and the evaluation were clear. However, there are some issues: (1) could you specify how the masks were obtained? were they generated automatically or manually?; (2) in Table 1, the std were 0 for some SSIM and MSE values? could you please explain this?; (3) what are the downstream applications for your task? Additionally, could you also clarify the details of "The dataset will be made available for sharing after the final data acquisition." or clarify when your dataset will be released?

---

### Decision · Program_Chairs · 2024-04-26

Accept